# Sensing Forests Directly: The Power of Permanent Plots

**DOI:** 10.3390/plants12213710

**Published:** 2023-10-28

**Authors:** Oliver L. Phillips

**Affiliations:** School of Geography, University of Leeds, Leeds LS2 9JT, UK; o.phillips@leeds.ac.uk

**Keywords:** networks, tropical, forests, people, nature, climate, biodiversity, global change, remote sensing, equity

## Abstract

The need to measure, monitor, and understand our living planet is greater than ever. Yet, while many technologies are applied to tackle this need, one developed in the 19th century is transforming tropical ecology. Permanent plots, in which forests are directly sensed tree-by-tree and species-by-species, already provide a global public good. They could make greater contributions still by unlocking our potential to understand future ecological change, as the more that computational and remote technologies are deployed the greater the need to ground them with direct observations and the physical, nature-based skills of those who make them. To achieve this requires building profound connections with forests and disadvantaged communities and sustaining these over time. Many of the greatest needs and opportunities in tropical forest science are therefore not to be found in space or in silico, but in vivo, with the people, places and plots who experience nature directly. These are fundamental to understanding the health, predicting the future, and exploring the potential of Earth’s richest ecosystems. Now is the time to invest in the tropical field research communities who make so much possible.

## 1. Introduction

We live in a time of extraordinary and continuous technological change that is revolutionizing how all science is conducted. Few domains are affected more than the study of forests and other ecosystems. With new eyes-in-the-sky on satellites, planes, and drones, we aim to probe the structure and metabolism of forests as never before. We are spending billions of dollars of public and private funds on lasers, radars, and other high-resolution sensors. Enabled by these and powered by cloud-computing and deep-learning technologies, we may be entering a new era of ecological discovery. At the same time, the Earth itself is changing as never before, so the need to measure these changes, understand them, and mitigate the increasing threats from climate change and nature depletion is greater than ever.

Opportunity and risk go hand-in-hand. Digital approaches certainly allow us to perceive nature in new ways, see new patterns, and make new connections, but they do not, per se, allow us to think any better. In some circumstances they may even blind us. Constant change in technologies, sensors, and platforms risks making us worse, not better, at perceiving and understanding long-term ecological change. More profoundly, our most important and deepest relationships with nature are not intellectual, but emotional and physical. The more we directly interact with nature, the happier and healthier we are and the more deeply we see it [1]. Those privileged to have grown up experiencing nature directly know this intuitively. Only by interacting directly with nature will we perceive it deeply and be open to the possibility of enchantment.

There is therefore a clear and present danger. The more we replace our own sense of nature with indirect, digital substitutes the weaker our connections and the less we care. And the less we love, the more we seek only to control. Ultimately, our science risks becoming as disenchanted and nature-depleted as the wider societies that we inhabit.

How can we unlock the scientific potential of technological innovation while avoiding the corrosive damage that neglecting sense-based skills and experience causes? How will we ensure that technologies work for the wider good, not only for the powerful owners and users of new tech? Since ecological wisdom will not come from machines, I believe we must cherish direct experience of nature and associated fundamental skills much more than we are. This is critical not simply for realising the potential of remote approaches for observing nature, but also for nurturing our own nature-based humanity. *In sum, to do “good” science we need to ensure that we scientists and those we train know how to sense forests directly*.

One technology offers exactly the kind of tree-centred, direct sensory interactions that our science needs. Not only has it proven vital for understanding forests and forest change, but it can help us even more. For almost two hundred years now, forests have been studied by mapping, measuring, and identifying all the trees in defined patches of land (“plots”), and carefully following each one’s individual fate. Familiarity, perhaps, breeds contempt. Yet this down-to-earth tool of forest science, now enriched by botanists, sharpened with standardised protocols and linked by international networks, is critical to the success of our new era of planetary ecology. Without plot networks, much of the potential scientific and societal value of the new tech will go unrealised. With them, the synergies that stand to be unleashed can be widely transformational and beneficial. 

### The Nature of Plots

Plot protocols were first established to inventory European forests and were applied in the tropics in the 19th century by the German forester Brandis, working for the British in Burma (now Myanmar) and India primarily to understand stocks of timber species [2]. Fundamental are standardized measurements of tree diameter, condition, and the identification of species. When censuses are repeated over time, they generate precise records of tree growth rates, recruitment into the population, and mortality, including when and sometimes how individual trees died. The longer the monitoring, the greater the value. Complementary measurements are often made of tree location, height, canopy condition, of lianas, as well as site topography and soil physical and chemical conditions. In tropical forests, due to their high species diversity, identification usually requires collections of botanical vouchers and positive species-level identifications in the field and in the herbarium by comparing to validated reference collections. 

This work is sometimes perceived as being simple, even simplistic. Nothing could be further from the truth. Few amongst us can measure or collect a large tree tropical tree correctly. Fewer still are able to identify them reliably when the local flora often runs to thousands of tree species alone. Many of the most undisturbed sites in the Amazon and Congo forests are especially remote. Simply reaching them is expensive and often risky; sustaining them more so. Once there, a lack of reliable electricity is one among many possible logistical and physical challenges encountered. 

With data so hard-won, teamwork is essential. Critical is combining specialists in tree measurement with trained climbers and botanists able to collect and identify the many tree species present. Vital post-field work includes transport and drying of hundreds of samples, making voucher collections, imaging them, curating them, and of course identifying them. Each species has its unique phenology and hence constantly changing canopy reflectance. By involving indigenous participants, and considering plant stem characters, DNA samples, isotopes, fungal and animal symbioses, soil, foliar and architectural variation, and within-individual ontological change, we can improve identification and enrich our science immeasurably. *In short, the challenge and complexity of tropical biodiversity is underestimated from afar, and only knowable with hands-on community engagement and attention.*

Combining multi-site efforts by different teams increases the range of questions we can ask and the power of analyses. Multi-institution networks have been developed to support this. These include regional networks aiming to inventory and monitor forests in areas often many times the size of most European countries, and international networks that seek to combine, support, and connect colleagues’ efforts across whole biomes or continents. While there are dozens of plot-based networks today with many different geographic or scientific foci, this is still a recent development. The first international plot network in South America, RAINFOR, coalesced at the turn of the millennium. ForestPlots.net, the first initiative to unite multiple networks and support their researchers within situ data management, began a decade later. Now, as researchers, institutions, and networks work increasingly closely, the mantra “better together” is underpinning a wave of new science and hyper-collaboration. While major challenges remain concerning equity and sustainability (cf. for example, [3]), forest plots and their networks have truly transformed tropical ecological science. In the 2020s, the pace and connectivity of international plot-based research is accelerating and being augmented by new techniques in which forest plots provide the critical infrastructure and baseline knowledge to support complementary approaches. In particular, the opportunities to connect bottom-up and top-down views of forests are richer than ever.

## 2. Achievements and Contributions

To outline the impacts and potential of tropical plots I focus on high-level domains in forest science. Within each, themes and landmark papers are selected to illustrate the irreplaceable impact of plots. Finally, I sketch out some domain examples of how plots can unlock potential of other approaches too. 

This brief overview, incomplete and shaped by personal experience as it is, demonstrates multiple irreplaceable contributions to almost all aspects of our science. Table 1 lists examples of scientific breakthroughs that depend on long-term tree-by-tree measurements on the ground. Among the forest science domains I highlight are community-based and species-centric approaches to ecology. For example, plots revealed where the most diverse forests on Earth are, why, and what exactly they are made of. With them, we have also addressed key ecosystem science questions, such as which forests are most productive, why, and what happens to carbon after plants fix it from the air. Plots also help us materially to manage forests more sustainably for their products. In addition—suitably so given that it is people that make plot measurements and identifications—plot-based analyses shine deeper light on humanity. When combined with anthropological and ethnobotanical approaches, plots show us not only which species and forests people value most, and why, but can change our understanding of what tropical nature is and how humanity has long been part of it. 

Perhaps most salient of all for our planet in peril are the contributions of direct, long-term tropical forest observations in plots to many of the key themes of what we call “global change forest science”. This includes showing where carbon stocks and flows are concentrated geographically, including below-ground as well as above it, and why, and by which species, and what are the risks to these, and where carbon and biodiversity attributes and processes are changing, and how and why climate change is driving some of these fundamental changes and responding to them. Combined with models of climate, topography and soil, plots give us unique insight into where forests are at greatest risk of change, and conversely might resist the threat of heat and drought long after model approaches can suggest irreversible tipping-points. 

In all, a wide range of scientific breakthroughs required long-term, tree-by-tree, species-by-species measurements in plots. The Table 1 examples include where plots have enriched or complemented measurements by other approaches; for most of these, plots were or are critical. All the phenomena are either invisible from space or only interpretable with the help of ground measurements. Single-census plots may adequately sense some target properties (e.g., composition, structure), but the extra rigour associated with “permanent” plots adds value even when the plot is used for snapshots of forest state properties. Permanent plots ensure more plant and soil collecting, higher identification quality, better tree measurements, and so on.

The majority of examples in Table 1 evidence advances that plots have not simply enabled (i.e., discovered with plots, unknowable without them) but for which they were almost uniquely essential for. The plot-based methodological package delivered the science advance, sometimes with associated laboratory support such as via soil nutrient analyses or herbaria plant identification. Other publications exemplified the critical role of plots in calibrating or validating inferences from other measurement methods or modelling approaches. 

A key application in recent years has been attempts to calibrate and validate remote-sensing techniques, especially for those that aim to map forest biomass. The relationship between the ground- and remote-measurement communities is, however, intrinsically problematic. There is huge asymmetry and global disparity in the investments made for space (large, mostly North) compared to those on land (small, mostly Global South). But here, also, lie opportunities for more integrated and more equitable science. Remote sensing techniques permit comprehensive mapping of forest area and disturbance at high spatial and temporal resolutions, but many ecological structure, function, biodiversity and change attributes are less well perceived from above. Soils, for example, are largely invisible from space, and most forest plant and animal species are below the top canopy, so are unseen. Space-borne LiDAR estimates of Amazon forest biomass are spatially biased by missing major large-scale gradients in tree species composition—but plots measure species, so the best maps incorporate ground-sensed wood density and tree allometry [43,44]. Over the coming decade, plots will be essential for calibrating and validating remote estimates of biomass change and dynamics too, as is already the case for spatial variation in forest structure and canopy dynamics [45,46] and other aspects of forests including biodiversity and biodiversity changes.

The core contribution of remote sensing is to provide synoptic views of many forest states and functions. The natural complementarity between organism-centred, ground-based local measurements and top-down mapping capacities is so obvious that the past failures to properly integrate ground-based measurements into satellite ‘missions to planet Earth’, collectively costing billions of dollars, represents a serious misuse of public funds. In Section 4, below, I discuss current initiatives attempting to correct this error.

## 3. Threats Faced

Our society’s fascination with what is new drives technological innovation. However, innovation is not sufficient for a healthy society, nor is it a good measure for deciding scientific priorities in a world that needs continuity and long-term evaluation. New sensors, for example, may ‘disruptively’ measure a physical canopy property very precisely, but are useless for monitoring forest change unless they persist for decades. Which they rarely do. Human skills and direct observation will remain essential for long-term monitoring and critical for building deep scientific understanding. Therefore, without some healthy scepticism and sustained ground-control investment, technological innovation can even undermine our long-term vision, making us blinder.

Meanwhile we fund most fieldwork as quick-hit curiosity-led research. But tying individual censuses to short-term hypothesis-testing research funding is no way to run a long-term experiment, let alone a network. Really important and revolutionary work often emerges from doing the same thing again and again, combining careful observations over space and time, and the long-term outcomes may not even be foreseeable. It is a matter of some curiosity that few funding agencies have ever figured this out.

Just as people need plots, plots need people. Without new censuses and skilled technicians, they simply die. There are hundreds of “permanent” plots across the tropics that we are losing right now. While satellites have big budget space agencies and corporate sponsors who place them in orbit, the acquisition of research-grade, botanically identified permanent plot tropical ground observations depends on individual investigators and short-term grants. There is no technological alternative to skilled experts carefully identifying and monitoring forests tree-by-tree. Hence, because of our tendency to deify technology, obsess with the new, and the short-termism of science, acquiring and processing high-quality plot data and sustaining the people who generate them is a permanently challenging task.

## 4. Tackling the Challenges, Unleashing the Opportunities

If securing long-term funding is the perpetual challenge for direct forest monitoring, others compound it. Few botanists can identify all trees in plots, our cultural biases undervalue repeated field work, and political conflict erodes nations’ abilities to invest in science and to protect nature. In the face of these, researchers need to join forces, when possible, to make scientific progress and overcome the practical challenges they face.

The sustained involvement of national and international networks of researchers can help provide critical mass, training, equitable data sharing, and other opportunities to counteract some of these challenges [3]. Elsewhere, colleagues and I have traced the trajectory and impact of recent network building in detail (e.g., [47,48,49]). These networks have completely transformed tropical ecology and continue to do so. Without the fanfare of launching billion-dollar missions to Planet Earth, many of the cutting-edge discoveries, advances in fundamental knowledge, and understanding of complex global change processes in tropical ecology are delivered by plot-networked science (examples in Table 1). Recent developments have included connecting these networks to one another. ForestGEO, for example, is supporting multi-network approaches to training in data analytical skills [48], bringing together young scientists from the global tropics to invest in these skills and develop common approaches. ForestPlots.net has evolved into a meta-network (Figure 1), supporting two dozen networks and more than two thousand partners with the tools to manage and analyse their data, and to decide and control how, with whom, where, and when to share them [49]. 

The networking and meta-networking initiatives represent relatively low cost interventions. But what could be more valuable than sustaining what works? Ensuring continuity of observations is our only defence against the creeping phenomenon of shifting baselines and the danger it represents. How blind we would be if the humble thermometer was not invented in the 17th century and widely distributed ever since. No amount of new sensors can replace the fundamental insights gained from repeating standardized measurements for a long time in many places.

In parallel with the networking revolution, scientific recognition is growing that ground plots are more than simply ‘forest inventories’ but are the fundamental tool of tropical forest ecology and many of its key applications. As well as the examples explored above in Section 2, it is now recognised that they are essential for mapping forests [14] and core to all forest biomass assessment [13,51]. Permanent plots with measured, identified trees provide the framework around which other ‘ground’ tech, including terrestrial, drone, or airborne laser scanning, can be applied, multiplying their value. An example of how this is beginning to happen at scale is the GEO-TREES initiative [16]. Founded by a coalition of tropical forest networks and remote-sensing colleagues, this explicitly recognises that supporting ground measurements and the people who make them is critical for mapping and tracking Earth’s forest carbon. Building on the strengths of forest plot networks, GEO-TREES will support high quality ground data from long-term sites. When combined with complementary techniques (ground laser scanning of tree volume and airborne scanning of canopy structure), hundreds of sites have the potential to be biomass reference sites for multiple missions for decades to come. Critical to the success of GEO-TREES - and therefore to the space missions - is funding for fieldwork, training, data management, integration, and local, national, and international administration and leadership. This is estimated at under USD 100M in current costs. Not cheap, but less than 5% of space agency investments in remote sensors of forest properties.

If supported in full GEO-TREES will help map and track Earth’s forest carbon better. But will it make a lasting difference on the ground, or simply deliver data more efficiently to already privileged analysts, modelers, and technocrats? In spite of their remarkable contributions, field researchers and ground networks still remain marginalised and peripheral to global discourses. Clearly, deeper change is needed to realise the full value of permanent plot approaches to measuring, monitoring, and managing tropical ecosystems. True sustainability will involve investing both in the direct-observation ground plots and the skills, long-term careers, and motivation of tropical scientists. Only this will generate the needed long-term resilience and local capacity for global observations.

## 5. Recommendations and Conclusions

Permanent plots and associated human capacity can be mobilized to help address some of the leading global ecological challenges of our times. These include *diagnosing change* (e.g., where and why is carbon and biodiversity being lost on the Earth’s surface), *predicting the future* (e.g., which ecosystems will lose carbon from rising temperatures or increasing drought?), and *mitigating and adapting* (e.g., what ‘natural climate solutions’ will work, where, and with what species). This is a tall order though; to do this requires at heart a shift in our values. 

The key change that is needed, I believe, is to acknowledge that the future of tropical ecosystem monitoring depends on the future of the people doing it. Models and remote sensors need measures of trees, species, genes, or soils to compare with, but effective, collective, and permanent global forest observation requires going beyond field teams simply providing data to modelers and space agencies in the U.S., Europe, and Asia. We need a new deal for plots and their people, one which deeply values grassroots continuity and the people, processes, and places who enable it. Simultaneously we must make five big, connected, operational changes:

**1. *Match recognition of what ground communities provide) with massive changes in funding***: serious investment in the careers, continuity, and facilities of those working on the ground. Focusing foremost on people will deliver more equitable, sustainable, and effective global observation than the current techno-centric paradigm.

**2.** This will lead to better opportunities for ***integrated global observation***. Long-term ground records can anchor and connect the diversity of remote observation sensors and techniques that are all subject to obsolescence. Permanent plots can enable the world’s long-term scientific baseline and reference system not simply for biomass (e.g., 16, 23), but also for carbon fluxes, forest dynamics, and change, as well as for biodiversity. 

**3**. Centring this on hundreds of sites with research-grade, permanent plots across the tropics will enable ***an integrated early-warning system aided by permanent plots***. We could be diagnosing the global dynamics of mature tropical forests regularly and frequently from the ground and from above. We clearly still need new plots in less-sampled regions (c.f. Figure 1), but the biggest science priority is to ensure long-term continuity for existing plots, so that they really are *permanent*. 

**4.** This new capacity (1–3) will help reliably ***diagnose climate and CO_2_ sensitivities and integrate them into models and policies***, grounded in multi-biome, community, tree-level measurements. Ground observational networks will also be able to test and improve these predictions as the future unfolds. 

**5.** Last but not least, the better the quality, continuity, and diversity of tree-by-tree plot-data, the more we can use them to ***devise and evaluate realistic natural climate solutions***. This includes quantifying the global climate benefits that flow from indigenous land-titling, assessing what species can be planted where, and demonstrating how protected area networks conserve forest biodiversity and carbon in the face of climate change. 

Above all, now is the moment to value the tropical field research communities. They are fundamental to understanding the health, predicting the future, and exploring the potential of the world’s most diverse ecosystems. With national and international networks of permanent plots and their researchers providing a global public good, it is high time we treated them as such.

## Figures and Tables

**Figure 1 plants-12-03710-f001:**
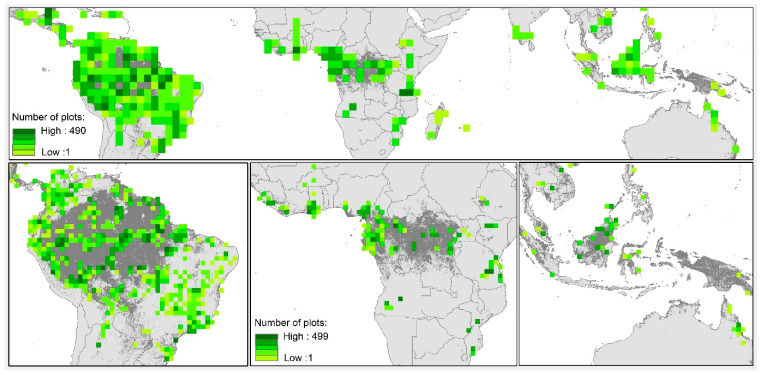
**A pan-tropical meta-network of forest plots and people**. Current extent of sample plots hosted by ForestPlots.net (map by Georgia Pickavance), with 7033 single- and multi-inventory plots contributing to 25 different national and international networks. In each, people identify and measure every tree greater than a threshold diameter (typically 10 cm, but often smaller) and, in most, the size and fate of each tree is tracked over time. Data-owners worldwide store, curate, and actively manage their data, while choosing how, with whom, and when to share or initiate their own collaborative work. More than 300 plots have been monitored for more than two decades, but the median length of all tropical monitored plots is only 10 years, demonstrating the great need to invest in continuity. **Top**: Pantropical plot sampling density per 2.5 degree square. Forest cover based on the Global Land Cover 2000 database with tree cover categories: broad-leaved evergreen; mixed leaf type; and regularly flooded. **Bottom**: The same with ForestPlots.net sampling displayed at higher-resolution (1-degree grid cells) for South America, Africa, and Southeast Asia and Australia. While forests are unevenly sampled, most of the climatic and geographic space across the humid tropics is represented [49] and some less sampled regions here are better covered by complementary networks (e.g., African woodlands by SEOSAW [50]; East Asian forests by ForestGEO [48]).

**Table 1 plants-12-03710-t001:** The power of plots: detecting, measuring and understanding forest ecology and change.

Forest Science Domain	Theme	Examples	Plot Criticality
** *Composition* **	Understanding local and regional floristic variation	Amazon community floristics and its drivers [4]	Essential: discovered with plots, unknowable without them
	Understanding floristic variation within and across biomes	Neotropical dry forest species and differentiation [5]	Essential: discovered with plots, unknowable without them
** *Diversity* **	Understanding variation in species richness, diversity and dominance	North-west Amazon and Andean forests are the global epicentre of arboreal diversity [6,7]	Essential: discovered with plots, unknowable without them
		Revealing 15,000 tree species in Amazonia [8]	Essential: discovered with plots, unknowable without them
		Predicting 73,000 tree species worldwide [9]	Essential: discovered with plots, unknowable without them
** *Ecosystem Processes* **	Productivity	Primary productivity and its large-scale climate and edaphic controls [10]	Essential complement: independent, direct bottom-up measure
	Respiration, Allocation	Tracking C fluxes and photosynthate allocation within ecosystems [11]	Essential: discovered with plots, unknowable without them
** *Biomass Carbon* **	Estimating and Mapping biomass	Species composition controls local to continent-wide biomass via taxon-dependent wood density [12,13]	Essential: impact of species on forest AGB is discovered with plots, unknowable without them
		Global biomass mapping with radar and airborne LiDAR needs plots (diameter, volume, species) [14,15,16]	Complements: parameterise or validate Earth Observation-informed modelling
** *Buried Carbon* **	Mapping carbon hotspots	Quantifying Congo Basin peatland carbon [17]	Complements: validation of EO-informed modelling
** *Forest Peoples’ Cultural Influence* **	Understanding where and how indigenous people managed forests	Legacies of indigenous forest domestication and management in Amazonia [18]	Essential: discovered with plots, unknowable without them
** *Soils* **	Revealing how soils drive forest ecology	Soil physical and chemical conditions control forest biomass, productivity and dynamics [19]	Essential: discovered with plots, unknowable without them
		Topography and water table depth controls on forest ecology [20]	Complements: provides long-term ecology to compare with remote-sensing
	Soil interactions with climate and biota	Climate-sensitive mycorrhizal impacts on global forest ecology [21]	Essential: discovered with plots, unknowable without them
** *Forest Change and Global Change Drivers* **	Changing forest structure and carbon	Discovering the carbon sink in mature forests [22]	Essential: discovered with plots
		Measuring change within intact forests [23,24,25]	Essential: measured with plots, largely invisible from space
	Changing forest dynamics	Baseline and change in Amazon forest growth, recruitment, mortality, residence times [26]	Essential: discovered with plots, largely invisible from space
	Attributing drivers of dynamic changes	Attributing climate, CO_2_ and residence time controls of continental changes in biomass, growth and mortality [27]	Essential: discovered with plots, largely invisible from space
	Changing forest diversity and composition	Thermophilization of Andean forests [28]	Essential: discovered with plots, unknowable without them
		Xerophilization of Amazon forest composition [29]	Essential: discovered with plots, unknowable without them
	Impacts of extreme drought events	Drought and thermal sensitivity of forest growth, mortality, biomass [30,31]	Essential: discovered with plots, unknowable without them
	Predicting climate-change induced future forest change	Long-term climate sensitivity of tropical forests [32]	Essential: predicted with plots, provides ground constraints for dynamic climate-vegetation models
	Defaunation impacts on forest demography and composition	Massive changes in tree species regeneration in “empty forests” [33]	Essential: measured with plots, invisible from space
** *Making Models of Nature* **	Initiating and Calibrating Models	Predicting future defaunation-induced carbon losses when large vertebrate fruit-dispersers are removed [34]	Essential: plots predict and constrain models of past and future changes which are invisible from space
		Establishing robust individual- and trait-based models of forest function [35]	Essential: provides in situ traits and long-term, species- and stand-level state, dynamics and change
		Establishing hydraulic-models of forest function [36]	Complements: provides ecosystem state, dynamics and change
	Validating models	Validating DGVM estimate of CO_2_-induced biomass gains in forests [37]	Complements: provide actual long-term stand-level state, dynamics and change
		Showing how variation in species’ hydraulic traits affects the long-term carbon balance of forests [38]	Essential: provides in situ trait measurements and long-term biomass growth, mortality, dynamics records
** *Managing Forests* **	Characterizing key species	Determining the diversity, abundance, frequency, distribution and vulnerability of timber tree species [39]	Essential: provides long-term, species abundance, frequency, distribution, reproduction across forest domain
	Improving sustainability	Establishing sustainable logging limits and size-class thresholds for forests [40]	Essential: direct validation of which management strategies work, which don’t
	Biodiversity Recovery	Revealing how species richness recovers fast but species composition very slowly in secondary forests [41]	Essential: long-term, ground-measured biodiversity and composition only possible via ground ID
	Carbon Sequestration	Establishing IPCC Tier I defaults for nation states to estimate their carbon uptake in secondary forests and intact forest growth [42]	Essential: long-term, ground-measured biomass changes and forest management

## Data Availability

Plot data underlying work introduced here are partly available from the relevant journal websites, some also as data packages at https://forestplots.net/en/publications#data.

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
