# Peer review of "Sensing Forests Directly: The Power of Permanent Plots"

_plants, 2023, doi:10.3390/plants12213710_

Round 1

Reviewer 1 Report

This is an eloquent piece clearly and unashamedly outlining the importance of permanent sampling plots in tropical forest ecology and particularly how they are needed to support new approaches. Indeed, the value of permanent plots is arguably greater than other approaches when considered as a network of internationally connected scientists as the author clearly elucidates. Overall, the message of the piece is clear and supported by appropriate evidence and references. There are a few sections that could do with a little more information/clarity and a few minor points to pick up on:

First off, I wonder if the title would be better as ‘Sensing Forests Directly: the Power of Permanent Monitoring Plots’? This gives a little more context to the title for those less familiar with the topic.

The second paragraph is important is it links fieldwork and nature appreciation. I don’t dispute this point but it would be good to provide a couple of references for support/further reading.

It would be valuable to provide some more details on the requirements for the long-term support for herbaria and the skills found there, particularly for field ID skills, and also the role of parataxonomists. Perhaps more than half of the work associated with a plot is the actual tree identification which, as the author knows, is not a trivial task so this needs a little more appreciation. In many countries, in my experience, national support for herbaria is strong (arguably better than the UK!), but how can this be harnessed for permanent plot work?

The work in Table 1 is somewhat geographically biased, as is the data on the map. The author could make more of this and note that the longer periods of time a series of plots have been measured, the greater their value. It looks like the number of plots in South America may be approaching an order of magnitude greater than that in Asia. Can the author please expand on what could/should be done to increase this coverage in Asia, perhaps building on the lessons from South America and noting that expanded coverage and greater monitoring lengths will increase their value. On a slightly tangential point, It would be worth including at least one example from the 2ndFor network in the table.

Page 6. Last para: This could easily be misread as suggesting that national agencies don’t support permanent sampling plots. This is rarely true as many countries have some kind of national forest inventory but the point I think the author is trying to make is that these plots are rarely large enough, correctly established, or having sufficient trees identified to species-level to contribute to global monitoring efforts. Can this be clarified/expanded upon please.

Minor points:

Page 1, 3rd para: Colon at end of the first sentence to run on to the second?

Page 2, 4th para: Basin rather than Basins?

Table 1 row 9: Too many abbreviations

Page 6, 1st para: Missing ‘are’ before ‘almost’

Pages 7 & 8: Please provide references/links to ForestGEO, ForestPlots and GEO-TREES at first mention

Page 8: Spell out TLS, DLS, ALS in full

Page 8, 3rd para: Please provide a reference for the estimated costs

Author Response

Thank you for the very positive and helpful review, much appreciated!  In response to the request for a little more information, clarity and other minor points

  1. Title.  -> I also include one-off ground plots and single-time properties too (eg for structure, floristics), but I agree that even here the extra rigour often associated with 'permanent' plots helps (eg more plant collections, higher quality of IDs and of tree measurements). So, on reflection I take your suggestion (the 'power of permanent plots', but without the 'monitoring', with the alliteration). I also added brief text to explain this in the legend for Table 1.
  2. The second paragraph. ->Yes. I have rewritten this somewhat and included a reference.
  3. "More details on the requirements for the long-term support for herbaria and the skills found there, particularly for field ID skills, etc.  ->Yes. I added 'enriched by botanists' to introductory comments.  Section 1.1 already emphasizes ID and botanists. I added new text to emphasize even more the challenge and interest of such work 

    "Critical post-field work includes transport and drying of hundreds or thousands of samples, making voucher collections, imaging them, storing them, and of course identifying them. Local tree florulas in many tropical forests run into the thousands of species, each with unique phenology and hence changing canopy reflectance. Considering stem characters, DNA samples, fungal and animal symbioses, foliar and architectural variation, and within-individual ontological change additionally helps identification and enriches our science.  In short, the complexity and challenge of tropical biodiversity is always underestimated from afar, and only knowable by engaging with it hands-on."

  4. The author could make more of this and note that the longer periods of time a series of plots have been measured, the greater their value. ->Yes, I added a sentence to Fig 1 legend. I insert "The longer the monitoring the greater the value" in the Intro. I emphasize long-term continuity in the new Recommendations section.
  5. Geographic bias in Table and Fig 1 / the number of plots in South America may be approaching an order of magnitude greater than that in Asia. Can the author please expand on what could/should be done to increase this coverage in Asia, perhaps building on the lessons from South America and noting that expanded coverage and greater monitoring lengths will increase their value.   -> This is a good point. There is lots of uneven sampling of course. I try to address concisely, via another sentence to Fig 1 legend, and text about gap-filling in the Recommendations section.
  6.  On a slightly tangential point, It would be worth including at least one example from the 2ndFor network in the table.  ->  Thank you. DONE.  The Requena Suarez ref. had 2nd For contribution and leadership but I add another now {Rozeendaal 2019} and introduce a new, unique application of plots {measuring biodiversity recovery}
  7. Page 6. Last para: This could easily be misread as suggesting that national agencies don’t support permanent sampling plots. This is rarely true as many countries have some kind of national forest inventory but the point I think the author is trying to make is that these plots are rarely large enough, correctly established, or having sufficient trees identified to species-level to contribute to global monitoring efforts. Can this be clarified/expanded upon please.  ->Yes, I have amended text to try to clarify this

Minor points:

Page 1, 3rd para: Colon at end of the first sentence to run on to the second?

thank you. DONE

Page 2, 4th para: Basin rather than Basins?

thank you, changed to 'forests'.

Table 1 row 9: Too many abbreviations

DONE

Page 6, 1st para: Missing ‘are’ before ‘almost’

DONE

Pages 7 & 8: Please provide references/links to ForestGEO, ForestPlots and GEO-TREES at first mention

DONE

Page 8: Spell out TLS, DLS, ALS in full

DONE

Page 8, 3rd para: Please provide a reference for the estimated costs

DONE

Reviewer 2 Report

According to the structure and contribution, the work is not an original scientific work, but a review work. The topic is interesting, current and you emphasize its importance well, but it is not presented in the most adequate way. You yourself state that the paper is a short overview, incomplete and shaped by personal experience. I leave it to the editors to decide whether the work is suitable for publication in this respected journal. If the editors decide to publish it, it should be structurally better arranged and especially emphasize the conclusions and your own specific recommendations.

Author Response

I thank the reviewer for the helpful comments.

The paper was submitted as an evidence-based Opinion piece. So, in developing the perspective and argument I did not use the conventional format and structure of a original research article or of a comprehensive review of the whole topic.

Reflecting on the comments, I have revised text throughout, and now developed and titled a whole new section (Recommendations) at the end of the paper to emphasize the conclusions and bring some specific recommendations. I think this provides a more compelling and rounded message to the paper. Thank you for this.

Reviewer 3 Report

I do agree that plot data is critical in understanding the world's ecosystems both in initial measurements and in continuing sampling to assess change. However you have obviously not kept current in the advances in using various RS data to study ecosystems. You will never have enough resources to undertake the type or plot sampling your propose, especially given the remoteness, vastness of coverage and political instability. Rather than using phrases such as" deification of technology" and complaining about budgets of the earth observations missions, your article would be better served in proposing a plan to use both. The current and future measurements taken for plots  can greatly help in the sensor design and algorithms to produce the required monitoring of ecosystems.

" there are no national agencies investing in acquiring research-grade plot data"

Wrong.there are long term government supported monitoring programs:

1. The US Forest Service has been routinely collecting plot data for many years-The Forest Inventory and Analysis (FIA).

2. National Science Foundation Long Term Ecological Research Sites combine both traditional ecological studies with satellite and aircraft collected AVRISNextGen hyperspectreal data.

wrong check out Greg Asner's work using hyperspectral data.

no technological solution to doing sustained tree-by-tree identification and measurements.

Take a look at the NASA/JPL SHIFT airborne field campaign.

Author Response

I thank the reviewer for their comments.  I am no remote sensing specialist but I work with some who are and am familiar with many of the techniques' applications in tropical ecology. The paper is an opinion piece of course, with personal views. I reflect on society and science, including the values that drive both. Technological optimism and short-termism (crudely, our fascination with the new and shiny) deeply determine what we value, and in my view often blind us as much as help. I don't expect all to agree with every word but am entitled to question the underlying values which drive much of what we do.

While I develop a critique I do point out the complementarities and shared opportunities. Eg "The natural complementarity between organism-centred, ground-based local measurements (plots), and top-down mapping capacities are so obvious that the failures to properly integrate ground-based measurements into satellite ‘missions to planet Earth’ collectively costing billions of dollars represents a serious misuse of public funds. In section 4, below, I discuss current initiatives attempting to correct this error", and other locations beside. 'Integration' is a key priority in the new 'Recommendations'.

specific, brief responses

*agency investment in long term plots text.  -> I agree that my original text was  incorrect, as Rev 1 also pointed out. It lacked especially the 'tropical' context and relied too much on the phrase 'research grade'. FIA is an amazing programme. But no tropical nation has managed to come close - and especially so lacking in permanent plot monitoring, including high-quality identifications (which in the tropics always require difficult collections and identifications, with special skills), or regular exposure to the rigours of peer review and generating traceable public voucher collections for tree biodiversity. Several may be trying (eg Gabon has hundreds of Govt funded 1-ha PSPs now using RAINFOR protocols) but I am not aware of sustained successes yet. That text has been reworded.

*NSF-LTER - again, I agree great value, but with a few exceptions temperate I think.

*NASA/JPL SHIFT airborne field campaign. - looks interesting but the website I found concentrates on California and unclear to me to what extent/if at all it could generate reliable large-area species ids, especially for tropical floras which extend into the 1000s for trees alone. 

*Hyperspectral airborne work by Asner and others have impressive survey records in tropical forests, with estimates of canopy function and nutrition. There is great value here, but again it is not tracking species-level biodiversity over time I think.

All the above and more have unique qualities. One can envisage them being part of global tropical forest monitoring efforts, along with LiDAR, radar and other technologies. Many probably should. Lots of potential for synergy. But my point is that none replaces tropical forest PSPs in providing long-term demographic and floristic monitoring of TMFs, and each of those programmes likely outspends TMF ground efforts by a huge margin.  

Changes made include a careful re-evaluation and rewriting of sections of the manuscript. For example, a new Recommendations section has been added, and I used this to build more on the message of the advantages of integrating different approaches. Please see the final two paragraphs before section 2 which also have revised and expanded text.

Again, I don't wish to deny the core contribution of remote-sensing in providing synoptic views of many forest state and function variables. But the point of the opinion piece is to show how ground people, skills and processes provide much that is unique. This is frequently overlooked (or taken for granted) because our science and our society are skewed by a techno-centric approach to nature.

Round 2

Reviewer 2 Report

The paper has been supplemented and edited according to the recommendations and can be accepted in the category Opinion. As stated in the first revision, the work is current and contributes to a better understanding of the issues of research and monitoring of forest vegetation.

Author Response

I sincerely thank the reviewer for their positive support and their contribution in helping to improve the manuscript.

Much appreciated.

I have taken the opportunity to go through the whole again and edit for precision of meaning where I thought it would be helpful.

Reviewer 3 Report

I guess since it’s an option piece that the authors bias 

can be expressed. However, the author admits the lack of knowledge of remote sensing technology. Plot data will always be severely limited it sampling space, location, variety of ecosystems samples, & tine frame. The author discounts the value of RS measurements. I the a better argument is the combination of both approaches. We heavily rely on “ground truth” to understand was we are seeing in our data. I do believe in the need to have extensive long term plots. The author did make some changes, but I still think a more balanced approach would get his message across.

Author Response

Thank you again for the reviews.

I am certainly not a RS specialist but have a reasonable view of its potential (much) and limitations (some). 

As I learnt when writing this opinion, the thrust of my thinking in the intro is not at all unusual in a social science context. But, I think, these views about what we value and why are seldom expressed in our technology-led natural science world. They may make uncomfortable reading in places but I have worked hard to explain how I see it. And I repeatedly make absolutely clear that, of course, RS has much to offer especially if we work together better.  

for example... "Without plot networks much of the potential scientific and societal value of the new tech will go unrealised. With them, the synergies that stand to be unleashed can be widely transformational and beneficial."

I agree, a key question then is how?   I haven't tried to tackle this fully, but tried (prompted by this review and another) to emphasise the deeper value challenge (as I see it), as well as make some recommendations at the end for how plots can contribute further to tropical science including of course unlocking those bottom-up/top-down synergies.  There is also text addressing these synergies/opportunities with RS on page 6 and on page 8. 

I have now taken the opportunity to go through the whole once more with a fresh eye to review and change the language for clarity. As well as the more philosophical musings I aim to make clear the many contributions of direct sensing on the ground, the challenges faced by field communities, and include a strong case for more integration with RS approaches as a step toward better forest monitoring,

Round 3

Reviewer 3 Report

The corrects made by author significantly enhanced his message and the manuscript is ready for publication